# Time-resolved scattering of a single photon by a single atom

Victor Leong[1,2,†], Mathias Alexander Seidler[1], Matthias Steiner[1,2], Alessandro Cerè[1] & Christian Kurtsiefer[1,2]

Scattering of light by matter has been studied extensively in the past. Yet, the most fundamental process, the scattering of a single photon by a single atom, is largely unexplored. One prominent prediction of quantum optics is the deterministic absorption of a travelling photon by a single atom, provided the photon waveform matches spatially and temporally the time-reversed version of a spontaneously emitted photon. Here we experimentally address this prediction and investigate the influence of the photon's temporal profile on the scattering dynamics using a single trapped atom and heralded single photons. In a time-resolved measurement of atomic excitation we find a 56(11)% increase of the peak excitation by photons with an exponentially rising profile compared with a decaying one. However, the overall scattering probability remains unchanged within the experimental uncertainties. Our results demonstrate that envelope tailoring of single photons enables precise control of the photon–atom interaction.

[1] Centre for Quantum Technologies, National University of Singapore, 3 Science Drive 2, Singapore 117543, Singapore. [2] Department of Physics, National University of Singapore, 2 Science Drive 3, Singapore 117542, Singapore. † Present address: Data Storage Institute, Agency for Science, Technology and Research (A*STAR), Singapore 138634, Singapore. Correspondence and requests for materials should be addressed to C.K. (email: christian.kurtsiefer@gmail.com).

The efficient excitation of atoms by light is a prerequisite for many proposed quantum information protocols. Strong light-matter interaction by using either large ensembles of atoms[1,2] or single atoms inside cavities[3–5] has received much attention in the past. More recently, significant light-matter interaction has also been observed between single quantum systems and weak coherent fields in free space[6–11]. The time-reversal symmetry of Schroedinger's and Maxwell's equations suggests that the conditions for perfect absorption of an incident single photon by a single atom in free space can be found from the reversed process, the spontaneous emission of a photon from an atom prepared in an excited state[12–20]. There, the excited state population decays exponentially with a time constant given by the radiative lifetime $\tau_0$ of the excited state, and an outward-moving photon with the same temporal decay profile emerges in a spatial field mode corresponding to the atomic dipole transition[21]. Therefore, for efficient atomic excitation the incident photon should have an exponentially rising temporal envelope with a matching time constant $\tau_0$ and propagate in the atomic dipole mode towards the position of the atom[22].

For a more quantitative description of the scattering process we follow ref. 15, which assumes a stationary two-level atom interacting with a propagating single photon in the Weisskopf-Wigner approximation. The photon-atom interaction strength depends on the spatial overlap $\Lambda \in [0,1]$ of the atomic dipole mode with the propagating mode of the photon, where $\Lambda = 1$ corresponds to complete spatial mode overlap. We consider scattering of exponentially decaying and rising photons described by the probability amplitude $\xi(t)$

$$\xi_\downarrow(t) = \frac{1}{\sqrt{\tau_p}} \Theta(t) e^{-\frac{t}{2\tau_p}} \qquad (1)$$

and

$$\xi_\uparrow(t) = \frac{1}{\sqrt{\tau_p}} \Theta(-t) e^{\frac{t}{2\tau_p}}, \qquad (2)$$

where $\Theta(t)$ is the Heaviside step function and $\tau_p$ is the coherence time of the photon. Integrating the equations of motion in ref. 15 leads to analytic expressions for the time-dependent population $P_e(t)$ in the excited state of the atom for both photon shapes:

$$P_{e,\downarrow}(t) = \begin{cases} \frac{4\Lambda\tau_0\tau_p}{(\tau_0-\tau_p)^2} \Theta(t)\left(e^{-\frac{t}{2\tau_0}} - e^{-\frac{t}{2\tau_p}}\right)^2 & \text{for } \tau_p \neq \tau_0 \\ \frac{\Lambda t^2}{\tau_0^2} \Theta(t) e^{-\frac{t}{\tau_0}} & \text{for } \tau_p = \tau_0 \end{cases} \qquad (3)$$

and

$$P_{e,\uparrow}(t) = \frac{4\Lambda\tau_0\tau_p}{(\tau_p+\tau_0)^2} \left[ e^{\frac{t}{\tau_p}}\Theta(-t) + e^{-\frac{t}{\tau_0}}\Theta(t) \right]. \qquad (4)$$

In this work, we measure the atomic excited state population dynamics by scattering single photons with identical power spectrum, but different temporal envelopes. While the overall scattering probability only depends on the power spectrum, the dynamics depends on the temporal envelope of the incident photon. In particular, an exponentially rising envelope leads to a higher instantaneous excited state population than an exponentially decaying envelope.

## Results

**Experimental setup.** In our experiment (Fig. 1), we focus single probe photons onto a single atom, and infer the atomic excited state population $P_e(t)$ from photons arriving at the forward and backward detectors $D_f$ and $D_b$ (refs 23–25). We obtain $P_e(t)$ directly from the atomic fluorescence measured at the backward detector $D_b$ with the detection probability per unit time $R_b(t)$,

$$P_e(t) = \frac{\tau_0}{\eta_b} R_b(t) \qquad (5)$$

where $\eta_b$ is the collection efficiency. However, the detection rate in such an experiment is relatively small and therefore susceptible to detector noise. Alternatively, $P_e(t)$ can be determined from the detection rate at the forward detector $D_f$ with a better signal-to-noise ratio. The probability per unit time of detecting a photon in the forward direction at time $t$ is given by $R_{f,0}(t) = |\xi(t)|^2$ without an atom, and by $R_f(t) = \left| \xi(t) - \sqrt{\frac{\Lambda}{\tau_0}P_e(t)} \right|^2$ with an atom present. The atom alters the rate of transmitted photons via absorption and re-emission towards the forward detector $D_f$. Therefore, any change $\delta(t)$ of the forward detection rate is directly related to a change of the atomic population,

$$\delta(t) = R_{f,0}(t) - R_f(t). \qquad (6)$$

The excited state population $P_e(t)$ is then obtained by integrating a rate equation,

$$\dot{P}_e(t) = \delta(t) - \frac{(1-\Lambda)}{\tau_0} P_e(t), \qquad (7)$$

where the last term describes spontaneous emission into modes that do not overlap with the excitation mode.

A schematic of the experimental setup is shown in Fig. 2. A single $^{87}$Rb atom is trapped at the joint focus of an aspheric lens pair (AL; numerical aperture 0.55) with a far-off-resonant optical dipole trap (980 nm)[8]. After molasses cooling, the trapped atom is optically pumped into the $5\,S_{1/2}$, $F = 2$, $m_F = -2$ state. Probe photons are prepared by heralding on one photon of a time-correlated photon pair generated via four-wave-mixing (FWM) in a cloud of cold $^{87}$Rb atoms[26,27]. The relevant energy levels are depicted in Fig. 2b: two pump beams with wavelengths 795 and 762 nm excite the atoms from $5S_{1/2}$, $F = 2$ to $5D_{3/2}$, $F = 3$, and a subsequent ensemble-enhanced cascade decay gives rise to the time ordering necessary for obtaining exponential time envelopes[20,28,29]. Dichroic mirrors, interference filters and coupling into single mode fibres select photon pairs of wavelengths 776 nm (herald) and 780 nm (probe). Adjusting the atomic density of the atomic ensemble[27], we set the coherence time $\tau_p = 13.3(1)$ ns of the generated photons, corresponding to a spectral overlap with the atomic linewidth of $\sim 90\%$ (ref. 30). To control the temporal envelope of the probe photon, the heralding mode is coupled to a bandwidth-matched, asymmetric Fabry-Perot cavity. The cavity reflects the herald photons with a dispersive phase shift depending on the cavity resonance frequency. Tuning the cavity on resonance or far-off resonance (70 MHz) with respect to the centre frequency of the herald photon results in exponentially rising or decaying probe photons[20]. The FWM source alternates between a laser cooling interval of 140 μs, and a photon pair generation interval of 10 μs, during which we register on average 0.054 heralding events on avalanche photodetector $D_h$. The probe photons are guided to the single atom by a single mode fibre. The spatial excitation mode is then defined by the collimation lens at the output of the fibre

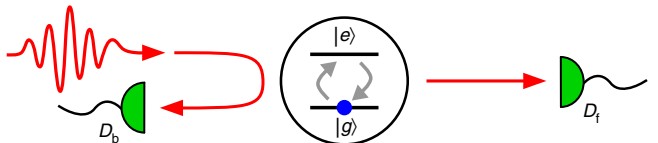

**Figure 1 | Single photon scattering by a two-level atom in free space.** The time evolution of the atomic excited state population is inferred by measuring photons in the forward or backward direction. $D_f$ and $D_b$: forward and backward detectors, $|g\rangle$ and $|e\rangle$: ground and excited levels of the atom.

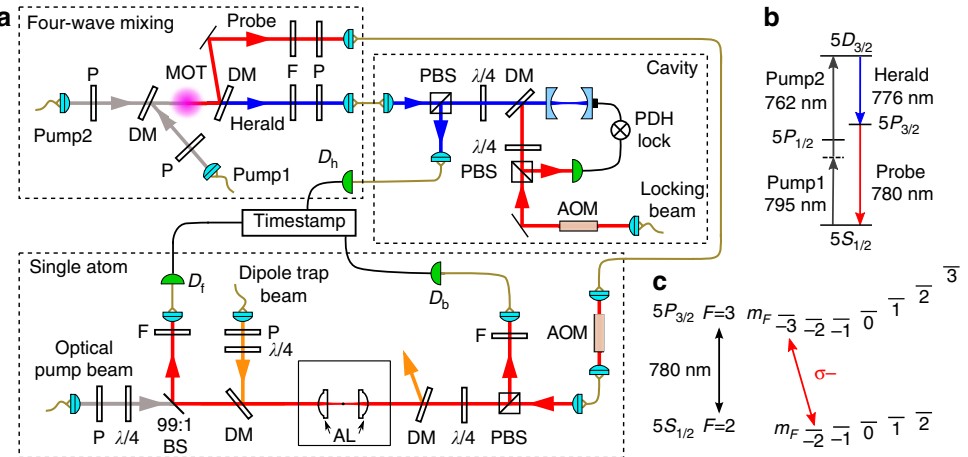

**Figure 2 | Experimental setup and level schemes. (a)** (Top left) Four-wave mixing part, providing heralded single photons: pump 1 (795 nm) and pump 2 (762 nm) are overlapped in a copropagating geometry inside the cold cloud of $^{87}$Rb atoms in a magneto-optical trap (MOT), generating pairs of herald (776 nm) and probe (780 nm) photons. The detection of a photon at $D_h$ heralds a probe photon. (Top right) Tuning the resonance of a bandwidth-matched cavity with respect to the heralding photon frequency controls the temporal envelope. (Bottom) Single atom part: A $^{87}$Rb atom is trapped at the focus of a confocal aspheric lens pair (AL; numerical aperture 0.55) with a far-off-resonant optical dipole trap (980 nm). The probe photons are guided to the single atom part by a single mode fibre and focused onto the atom by the first AL. Avalanche photodetectors $D_f$ and $D_b$ detect photons collected in forward and backward directions. An acousto-optic modulator (AOM) shifts the probe photon frequency to compensate for the shift of the atomic resonance frequency caused by the bias magnetic field and the dipole trap. $\lambda/2$, $\lambda/4$, half- and quarter-wave plates; $D_h$, $D_f$, $D_b$, avalanche photodetectors (APDs); DM, dichroic mirror; F, interference filter; PDH lock, Pound–Drever–Hall frequency lock electronics; P, polarizer; (P)BS, (polarizing) beam splitter. **(b)** Relevant level scheme of the four-wave mixing process in a cloud of $^{87}$Rb atoms. **(c)** Relevant level scheme of the single $^{87}$Rb atom in the dipole trap. The probe photons are resonant with the closed transition $|g\rangle = 5\,S_{1/2}, F = 2, m_F = -2$ to $|e\rangle = 5\,P_{3/2}, F = 3, m_F = -3$.

and the high numerical aperture AL. From the experimental geometry, we expect a spatial mode overlap of $\Lambda \approx 0.03$ with the atomic dipole mode[18]. The excitation mode is then collimated by a second aspheric lens, again coupled into a single-mode fibre, and sent to the forward detector $D_f$. A fraction of the photons scattered by the atom is collected in the backward direction, and similarly fibre-coupled and guided to detector $D_b$.

**Time-resolved transmission of single photons.** To investigate the dynamics of the scattering process, we record photoevent detection times at the forward detector $D_f$ with respect to heralding events at $D_h$. When no atom is trapped, we obtain the reference histograms $G_{f,0}(t_i)$ for exponentially decaying and rising probe photons, with time bins $t_i$ of width $\Delta t$ (Fig. 3, black circles). The observed histograms resemble closely the ideal asymmetric exponential envelopes, described by equations (1) and (2). The total probability of a coincidence event within a time interval of 114 ns ($\approx 8\,\tau_p$) is $\eta_f = 3.70(1) \times 10^{-3}$. When an atom is trapped, we record histograms $G_f(t_i)$ (Fig. 3, red diamonds). The two histograms $G_f(t_i)$ are very similar to the respective reference histograms $G_{f,0}(t_i)$. To reveal the scattering dynamics, we obtain the photon detection probabilities per unit time at the forward detector $R_f(t_i) = G_f(t_i)/(\eta_f \Delta t)$ with and without atom to use equations (6) and (7) to reconstruct the excited state population $P_e(t_i)$. Figure 4 shows the difference $\delta(t_i) = R_{f,0}(t_i) - R_f(t_i)$ for both photon envelopes, with mostly positive values. A positive value of $\delta(t_i)$ corresponds to net absorption, that is, a reduction of the number of detected photons during the time bin $t_i$ due to the interaction with the atom. For a photon with a decaying envelope, the absorption is close to zero at $t_i = 0$, and reaches a maximum at $t_i \approx 15$ ns, followed by a slow decay. In strong contrast, the absorption for photons with a rising envelope follows the exponential envelope of the photon, with a maximum absorption rate twice as high as that for photons with a decaying envelope.

We obtain analytical solutions for the expected differences in transmission $\delta(t)$ from equation (6) assuming the ideal photon envelopes equations (1)–(3). The magnitude and the dynamics of

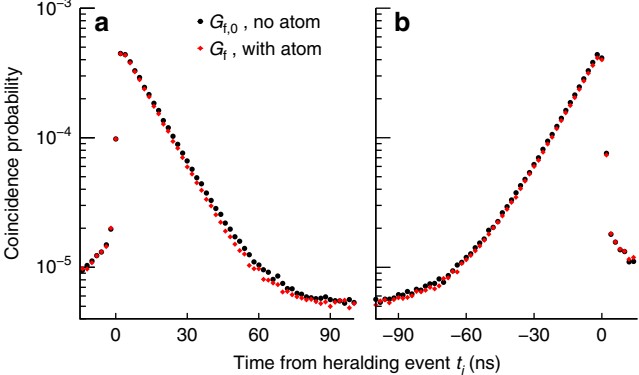

**Figure 3 | Time-resolved transmission of single photons.** Coincidence histograms between heralding detector $D_h$ and forward detector $D_f$ for exponentially decaying (**a**) and rising (**b**) probe photons, with a coherence time $\tau_p = 13.3\,(1)$ ns obtained from a fit to equations (1) and (2). Black circles: $G_{f,0}$, reference data taken without the trapped atom. Red diamonds: $G_f$, data taken with the atom present. The time bin size is $\Delta t = 2$ ns. Total measurement time is 1,500 h. Error bars indicate one standard deviation due to propagated Poissonian counting statistics and are smaller than the symbol size. We offset all detection times by 879 ns to account for delays introduced by electrical and optical lines.

the observed scattering are well reproduced for $\tau_p = 13.3$ ns and $\Lambda = 0.033$ (Fig. 4, solid lines). The observed peak absorption for the exponentially decaying photon is slightly higher than expected. We attribute this discrepancy to the imperfect photon envelopes that differ slightly from the ideal asymmetric exponential.

The interaction with the atom reduces the overall transmission into the forward detection path for both photon shapes. To quantify this behaviour, we calculate the extinction $\epsilon = \Delta t \sum_i \delta(t_i)$ by summing over the interval $-14\,\text{ns} \leq t_i \leq 100\,\text{ns}$ for exponentially decaying photons, and $-100\,\text{ns} \leq t_i \leq 14\,\text{ns}$ for exponentially rising photons, capturing almost the entire photon.

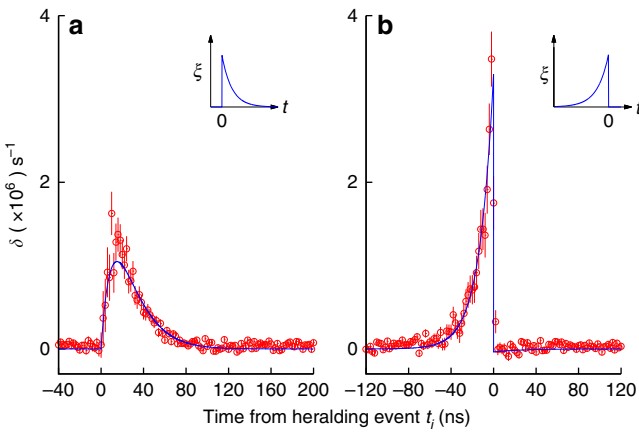

**Figure 4 | Time-resolved changes in single-photon transmission rates.**
Changes in the forward detection rates $\delta(t_i) = R_{f,0}(t_i) - R_f(t_i)$ induced by the interaction with the atom for exponentially decaying (**a**) and rising (**b**) probe photons. The time bin size is 2 ns. Solid lines: analytical solution of equation (6) using equations (1)–(3) for $\tau_p = 13.3$ ns, $\Lambda = 0.033$. Error bars represent 1 s.d., assuming Poissonian statistics for $R_{f,0}(t_i)$ and $R_f(t_i)$.

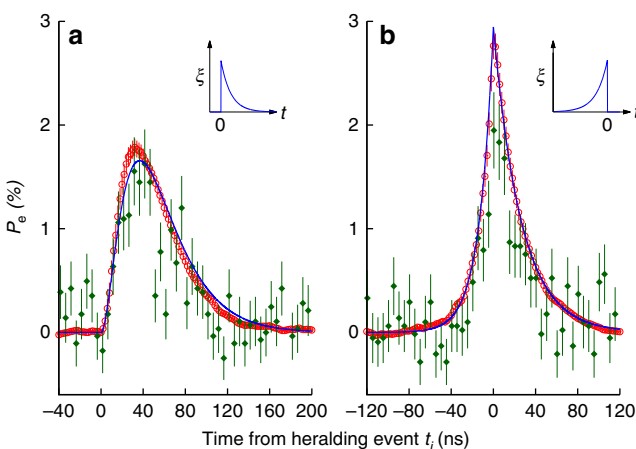

**Figure 5 | Time-resolved atomic excited state population $P_e$.**
(**a**) Exponentially decaying and (**b**) exponentially rising probe photons. Red open circles, time bin size 2 ns: $P_e(t_i)$ obtained from forward detection rates integrating equation (7). Error bars represent 1 s.d. of the distributions obtained by a Monte–Carlo method and assuming Poissonian statistics for the detection rates. Green filled diamonds, time bin size 5 ns: $P_e(t_i)$ obtained from the backward detection rates using equation (5). Error bars show 1 s.d. assuming Poissonian statistics for the detection rates. Solid lines: $P_e(t)$ from equations (3) and (4) using $\tau_p = 13.3$ ns, $\Lambda = 0.033$.

We obtain similar extinction values $\epsilon_\downarrow = 4.21\,(18)\%$ and $\epsilon_\uparrow = 4.40\,(20)\%$ for decaying and rising photons, respectively. The theoretical value of the extinction does not depend on whether the photon envelope is exponentially decaying or rising:

$$\epsilon = \int_{-\infty}^{+\infty} \delta(t)\,\mathrm{d}t = \Lambda(1-\Lambda)\frac{4\tau_p}{\tau_0 + \tau_p} \qquad (8)$$

For our parameters, $\tau_p = 13.3$ ns, $\Lambda = 0.033$, this expression leads to $\epsilon = 4.29\%$, which is close to our experimental results.

**Time-resolved atomic excitation probability.** The excitation probability $P_e(t_i)$ (Fig. 5, red circles) of the atom is obtained from the differences in the forward detection rates $\delta(t_i)$ and by numerically integrating equation (7). The exponentially decaying photon induces

a longer lasting but lower atomic excitation compared with the rising photon. We find good agreement with the analytical solutions given in equations (3) and (4) (Fig. 5, solid line). We do not observe perfect excitation of the atom from exponentially rising probe photons because of the small spatial mode overlap $\Lambda$. However, the peak excited state population for the exponentially rising $P_{e,max,\uparrow} = 2.77(12)\%$ is 56(11)% larger than for the decaying one $P_{e,max,\downarrow} = 1.78(9)\%$. The increase in the peak excitation $P_{e,\uparrow,max}/P_{e,\downarrow,max} = 78\%$ predicted by equations (3) and (4) for $\tau_p = 13.3$ ns, $\Lambda = 0.033$ is also in fair agreement with our findings.

The excited state population can also be directly determined from the atomic fluorescence, equation (5). To convert the coincidence histograms $G_b(t_i)$ between the heralding detector $D_h$ and backward detector $D_b$ into the excited state population $P_e(t_i)$ we have to account for the finite collection and detection efficiencies in the forward and backward path. For the backward path we independently measure the collection efficiency $\eta_b = 0.0126(5)$ and the detector quantum efficiency $\eta_q = 0.56(1)$. Figure 5 (green filled diamonds) shows the inferred excited state population $P_e(t_i) = R_b(t_i)/(\eta_b \Gamma_0) = G_b(t_i)/(\tilde{\eta}_q \eta_b \Gamma_0 \Delta t)$ with a time bin width of 5 ns, where $\tilde{\eta}_f = 0.0155(4)$ is the heralding efficiency in the forward path, corrected for the collection and detection efficiencies. Again, we find a qualitatively different transient atomic excitation for both photon shapes, in agreement with the theoretical model, but with worse detection statistics compared with the excited state reconstruction using the changes in the forward detection rates. The peak excitation probability and the signal rate can be improved by a larger spatial mode overlap $\Lambda$, which is currently limited by the numerical aperture of the focusing lens[31]. Other focusing geometries like parabolic mirrors can theoretically achieve complete mode matching $\Lambda = 1$ (ref. 11).

## Discussion

In summary, we have accurately measured the atomic excited state population during photon scattering and have demonstrated that the power spectrum of the incident photon is not enough to fully characterize the interaction. The exponentially rising and decaying photons have an identical Lorentzian power spectrum with a full-width-half-maximum $\Gamma_p = \frac{1}{\tau_p}$, but the transient atomic excitation differs. We have shown that the scattering dynamics depends on the envelope of the photon, in particular that an atom is indeed more efficiently excited by a photon with an exponentially rising temporal envelope compared with an exponentially decaying one. However, when integrated over a long time interval $\Delta t \gg \tau_0, \tau_p$ both photon shapes are equally likely to be scattered as shown by our measurement of the extinction $\epsilon$ The advantage of exciting single atoms with exponentially rising photons is a larger peak excitation probability within a narrower time interval. Such a synchronization can be beneficial to quantum networks.

Our experimental results also contribute to a longstanding discussion about differences between heralded and 'true' single photons[32–35]. The atomic excitation dynamics caused by heralded single photons matches well the one expected from 'true' single photon states in our theoretical model, and therefore support a realistic interpretation of photons prepared in a heralding process.

## Methods

**Heralded single-photon generation.** The two pump fields have orthogonal linear polarizations. The 795 nm pump laser is red-detuned by $-30$ MHz from the $5\,S_{1/2}$, $F = 2$ to $5\,P_{1/2}$, $F = 2$ transition to avoid incoherent scattering. The frequency of the 762 nm pump laser is set such that the two-photon transition from $5\,S_{1/2}$, $F = 2$ to $5\,D_{3/2}$, $F = 3$ is driven with a blue-detuning of 4 MHz. We can vary the coherence time $\tau_p$ of the generated photons by changing the optical density of the atomic ensemble. We choose $\tau_p = 13.3$ ns as a trade-off between matching the excited state lifetime of $\tau_0 = 26.2$ ns and having a high photon pair generation rate. Longer coherence times can be achieved at lower optical densities, but at the cost of lower photon pair generation rates.

The probe photons are guided to the single atom setup by a 230 m long optical fibre. An acousto-optic modulator (AOM) compensates for the 72 MHz shift of the atomic resonance frequency caused by the bias magnetic field (7 Gauss applied along the optical axis) and the dipole trap. The AOM also serves as an optical switch between the two parts of the experimental setup; once a herald photon is detected, the AOM is turned on for 600 ns. The optical and electrical delays are set such that the probe photon passes the AOM within this time interval. Before reaching the atom, the polarization of the probe photons is set to circular $\sigma^-$ by a polarizing beam splitter and a quarter-wave plate.

The Fabry–Pérot cavity used to control the temporal envelope has a length of 125 mm and a finesse of 103(5), resulting in a decay time $\tau_c = 13.6(5)$ ns. The reflectance of the in-coupling mirror and the second mirror are 0.943 and 0.9995 respectively. We use an auxiliary 780 nm laser to stabilize the cavity length using the Pound–Drever–Hall technique.

**Data acquisition and analysis.** Figure 3 shows the coincidence histograms without additional processing. For the quantitative analysis (Figs 4 and 5) we subtract the accidental coincidence rate from the histograms. The accidental coincidence rate is caused by background events in the photodetectors, and determined from the histograms by averaging the detected coincidences rate over 300 ns outside the interval used to analyse the scattering dynamics.

The total acquisition time for the experiment was 1,500 h, during which the average photon coherence time was $\tau_p = 13.3(1)$ ns and the heralding efficiency was $\eta_f = 3.70(1) \times 10^{-3}$. We check for slow drifts in $\tau_p$ and $\eta_f$ by analysing the histogram $G_{f,0}$ every 60 min for $\tau_p$ and 20 min for $\eta_f$. The distribution of $\tau_p$ is nearly Gaussian with a s.d. of 0.9 ns, most likely caused by slow drifts of the laser powers and the atomic density; the distribution of $\eta_f$ has a full-width-half-maximum of $6 \times 10^{-4}$. We alternated between the decaying and rising photon profiles every 20 min to ensure that the recorded coincidence histograms are not systematically biased by slow drifts in $\tau_p$ and $\eta_f$.

**Temporal photon envelope.** The coincidence histograms recorded without atom (Fig. 3 black circles) differ slightly from the ideal asymmetric exponential functions described in equations (1) and (2).

These deviations are well explained by the model we use to describe the effect of the cavity[20]. For the exponentially decaying photons, the main deviation is a small rising tail, caused by the finite cavity detuning of 70 MHz. For the exponentially rising photons, we observe a small decaying tail due to the bandwidth mismatch between cavity and photon, and cavity losses.

**Data availability.** The data that support the findings of this study are available from the corresponding author upon reasonable request.

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

## Acknowledgements

We thank Y. Wang for sharing her numerical simulation code, and B. Chng for helpful discussions. We acknowledge the support of this work by the Ministry of Education in Singapore (AcRF Tier 1) and the National Research Foundation, Prime Minister's office (partly under grant no NRF-CRP12-2013-03). M.S. acknowledges support by the Lee Kuan Yew Postdoctoral Fellowship.

## Author contributions

V.L., M.S.T. worked mostly on the single atom setup, M.S.E., A.C. mostly on the FWM source. V.L., M.S.T., A.C. carried out the data analysis; M.S.T., A.C. and C.K. conceived the experiment and provided basic experimental support. All authors participated in writing the paper.

## Additional information

**Competing financial interests:** The authors declare no competing financial interests.

