## [Peer Review File · Nature Communications]

Reviewers' comments:

Reviewer #1 (Remarks to the Author):

"Time-resolved scattering of a single photon by a single atom" by V. Leung, et. al.

This article presents the results of measuring the absorption of a single photon, whose temporal envelope is tailored to resemble an ideal decaying or rising exponential profile, by a single, trapped atom. This work is very painstaking and careful, requiring a long duration (months) to acquire sufficient signals for their quantitative analysis. The main result presented is that the excitation dynamics of a single atom by a single photon depends on the photon's temporal pulse envelope and not just on its power spectrum. Namely, that the maximum excitation probability is markedly different for photons with an exponentially rising temporal envelope compared to photons with an exponentially decaying envelope. This result is in fair agreement with the model of a photon absorption by a two level atom presented in this work.

The authors have demonstrated how to prepare and manipulate photons with exponentially rising and exponentially falling envelopes (many of the details of the different elements of the experimental apparatus are contained in the references cited in this article). One of the remarkable results of this work is the agreement between the simplified 2 level atomic model and the experimental results presented. The manuscript is well written: there is a clear motivation given, a concise description of the model explored, and a good presentation of the results and analysis. In my review of the work, there are only a few recommendations I have which may help improve the article:

a) In figure 3, the raw data are shown (The coincidence probability as a function of the time from the heralding event) for the exponentially rising and for the exponentially falling photon envelopes, with and without an atom trapped in the experimental chamber. It could be helpful to demonstrate how well the time-reversal of the envelope is accomplished experimentally by taking the two traces without an atom present, time-reversing one of them, and plotting the residuals of their difference. Alternatively, the predicted coincidence shape for the exponential rising and falling envelopes could be plotted with the observations to quantify the deviation from the model behavior.

b) The discrepancies between the model predictions and the data in Figures 4 and 5 merits a more in-depth discussion.

In Figure 4, the authors plot the differences between the coincidence probabilities with and without atom present for each tailored photon envelope. These data are fit using the model presented (Note: In the caption to Figure 4, the authors state the fit is to Eq. (2)-(5), although I believe these data are fit using Eq.(1) and to the expressions found in the text just prior to Eq. (5)). There is a notable discrepancy for the decaying envelope data and the model near the maximum ($t \approx 15$ ns). This discrepancy lies outside the error bars for this plot. By contrast, the model shows no such discrepancy for the rising envelope data set.

In Figure 5, the plots of the predicted and observed excitation probabilities as a function of time from the heralding event, once more (not surprisingly) demonstrate the same asymmetry between the model and the results. Again, this is outside the quoted error bars for the decaying envelope measurements compared to the model predictions.

The source of this discrepancy could lie in the non-ideal transfer function of the Fabry-Perot cavity used to control the photon temporal envelopes, the behavior of the AOM switching on/off, motion of the atom in the trap as opposed to being stationary, and/or due to the light interacting with the atom having non-ideal polarization, for example. That the discrepancy depends on the shape of the photon envelope is remarkable. I would appreciate like some small discussion of the potential sources of this effect. [Also see d) below].

c) The authors also say that "corrections for accidental coincidences were applied in the analyzed data shown in Fig. 4 and Fig. 5". I would appreciate a short explanation of these corrections here. The authors are taking the difference between two, almost equal traces to extract small, precise readings. The form of the correction and the method for correcting the data are relevant here.

d) In the text at the top of p.4, the authors state the percent differences between the excited state populations produced by the different shaped envelopes. The measured value shows a 56% larger peak excited state fraction as compared to the model prediction of 78% increase. This discrepancy should be commented upon in context with point b).

e) Finally, the authors state: "The advantage of using exponentially rising photons is, therefore, to excite atoms at well defined instants in time." I suggest changing this to a more specific statement such as: "The advantage of using exponentially rising photons to excite a single atom is that this provides a larger peak excitation rate which is more temporally localized."

Overall, I enjoyed reading this manuscript: The authors present the results of carefully executed and thorough work. With some of the minor adjustments I suggested above, I recommend publication of this work: The findings are of great interest to researchers in this field as well as to a general audience.

Reviewer #2 (Remarks to the Author):

The authors examine the excitation process of a single atom by a single photon. In particular, they investigate the influence of the shape of the wavepacket of the incoming photon, whether it is exponentially decaying or rising. The experimental results are valuable because some theoretical predictions had stated that efficient coupling of photons to atoms requires exponentially growing light fields to mimic the inverse of the spontaneous emission process.

The experiment is a continuation of a series of efforts in the lab of Prof. Kurtsiefer, where extinction of a focused laser beam by a single atom was studied. In 2013 they investigated the effect of the laser pulse shape on the excitation process. In 2014 they reported the production of single photons with exponentially rising profile. The current manuscript puts those two experiments together in a real tour de force effort, involving 1500 hours of data acquisition. The work would be a very nice addition to the literature in quantum optics. Therefore, I recommend this work for publication in Nature Communications once the following issues have been addressed:

1- The authors show convincingly the difference between the peak excitation probability for the cases of exponential rising and decaying pulses. They also state the important conclusion that when integrated over a long time interval both photon shapes are equally likely to be scattered. The latter statement should be included in the bold abstract (first paragraph) to avoid misleading/confusing the general reader about the necessity of pulse shaping when performing spectroscopy.

2- In the second paragraph, the authors write "More recently, significant light-matter interaction has also been observed between single quantum systems and weak coherent fields in free space [18-21]." The authors seem to have selected references to vacuum experiments. Since the physics does not change in any way when the excitation is done in a dielectric, they should also include references to G. Wrigge, et al, Nature Physics 4, 60 (2008) and Vamivakas, et al. Nano Lett. 7, 2892 (2007) where this type of work was first demonstrated on molecules and quantum dots.

3- In the last paragraph, the authors state "Our experimental results also contribute to a long-

standing discussion about differences between heralded and "true" single photons." They should cite the literature on this long-standing discussion.

Reviewer #3 (Remarks to the Author):

1. Summary

The authors present their experiment on scattering a single photon off a single atom. The temporal shape of the input photon is set to be either exponentially rising or falling, depending on the detuning of a cavity. They demonstrate that the excited state probability as a function of time is different for the two cases, and that a higher probability is reached for the rising mode. The integrated probability is however the same, as predicted by the theory.

2. General comments

In general, I found the manuscript interesting and nicely written. The topic of the manuscript should be of interest to the general physics community. The authors show a nice agreement between experiment and the theory. The results are trustworthy and the statistical analysis (errorbars are shown and numbers are displayed with uncertainties) is ok. In conclusion, I find the manuscript appropriate for Nature Communications and I recommend publication.

Detailed comments:

3. Spatial overlap Λ

Can you discuss how you can improve the value for Λ ? For a Gaussian spatial mode, what is the best overlap with the atomic dipole pattern? Are there any ways to experimentally produce an input photon with $\Lambda=1$?

4. Forward and backwards detectors:

Why do you have a better signal to noise ratio using the forward detector compared to the backwards detector? The atomic dipole emission should be the same in the forward and backwards directions, and I would therefore think that the number of detected photons (and thereby the signal to noise ratio) would be the same in the two directions.

5. y-label on Fig 3.

I am a bit confused about the y-label of figure 3. It says "Coincidence probability $\times 10^{-5}$ " and the y-axis goes from 10^{-5} to 10^{-3} . Does this mean that the actual probability goes from 10^{-10} to 10^{-8} ? To avoid confusion, I would suggest that you delete the " $\times 10^{-5}$ ", and just write out whatever the correct numbers are on the y-axis. Also, is it not clear to me how G_f and $G_{f,0}$ normalized? What is the relation between the value of G_f and the time-bin duration of 2 ns?

6. Equation (7)

I am a bit puzzled about Eq. (7). The function $\Lambda(1-\Lambda)$ is maximal for $\Lambda=0.5$. Why is this Λ optimal? I would expect that $\Lambda=1$ would give the biggest epsilon. However, I see that $\epsilon=0$ for both $\Lambda=0$ and $\Lambda=1$. Please explain. Also, how can one make $\epsilon=1$? I see that $\epsilon=1$ for $\tau_p \gg \tau_0$. Is there a simple physical explanation why this is optimal? I would expect that $\tau_p=\tau_0$ would be optimal according to your argument about time-reversal.

7. 1500 hours

On the one hand, it is impressive that you can run your experiment for such a long time. On the other hand, it is not so nice that you need so long time to get good statistics. I think that it would be nice if

you added some text in the methods section describing the main reasons for needing so long measurement time and give some suggestions for how you can improve your setup in order to do faster measurements.

8. Formatting of figure 2

Figure 2 has many details but is quite small. The figure would be more readable if it was larger and covered the full page width.

9. Formatting of Fig. 3, 4 and 5.

There is a problem with the formatting of Fig. 3, 4 and 5. The figure and the figure caption are partially on top of each other, making it hard to read the first two lines of each figure captions.

Reviewer #1:

This article presents the results of measuring the absorption of a single photon, whose temporal envelope is tailored to resemble an ideal decaying or rising exponential profile, by a single, trapped atom. This work is very painstaking and careful, requiring a long duration (months) to acquire sufficient signals for their quantitative analysis. The main result presented is that the excitation dynamics of a single atom by a single photon depends on the photon's temporal pulse envelope and not just on its power spectrum. Namely, that the maximum excitation probability is markedly different for photons with an exponentially rising temporal envelope compared to photons with an exponentially decaying envelope. This result is in fair agreement with the model of a photon absorption by a two level atom presented in this work. The authors have demonstrated how to prepare and manipulate photons with exponentially rising and exponentially falling envelopes (many of the details of the different elements of the experimental apparatus are contained in the references cited in this article). One of the remarkable results of this work is the agreement between the simplified 2 level atomic model and the experimental results presented. The manuscript is well written: there is a clear motivation given, a concise description of the model explored, and a good presentation of the results and analysis. In my review of the work, there are only a few recommendations I have which may help improve the article:

a) In figure 3, the raw data are shown (The coincidence probability as a function of the time from the heralding event) for the exponentially rising and for the exponentially falling photon envelopes, with and without an atom trapped in the experimental chamber. It could be helpful to demonstrate how well the time-reversal of the envelope is accomplished experimentally by taking the two traces without an atom present, time-reversing one of them, and plotting the residuals of their difference. Alternatively, the predicted coincidence shape for the exponential rising and falling envelopes could be plotted with the observations to quantify the deviation from the model behavior.

Reply: We find that the semi-log plot of the coincidence probabilities in figure 3 demonstrates the deviations of the experimentally achieved photon shapes compared to the ideal asymmetric exponentials in the best possible ways among the representations we tried, including the suggestion of the referee. However, we added a paragraph in the methods section where we describe what

causes the photon-shape imperfections to clarify this better.

Added at the end of the methods section:

Temporal photon envelope: The coincidence histograms recorded without atom (Fig. 3 black circles) differ slightly from the ideal asymmetric exponential functions described in Eq. 1. These deviations are well explained by the model we use to describe the effect of the cavity [12].

For the exponentially decaying photons, the main deviation is a small rising tail, caused by the finite cavity detuning of 70 MHz.

For the exponentially rising photons, we observe a small decaying tail due to the bandwidth mismatch between cavity and photon, and cavity losses.

b) The discrepancies between the model predictions and the data in Figures 4 and 5 merits a more in-depth discussion. In Figure 4, the authors plot the differences between the coincidence probabilities with and without atom present for each tailored photon envelope. These data are fit using the model presented (Note: In the caption to Figure 4, the authors state the fit is to Eq. (2)-(5), although I believe these data are fit using Eq.(1) and to the expressions found in the text just prior to Eq. (5)). There is a notable discrepancy for the decaying envelope data and the model near the maximum ($t \approx 15$ ns). This discrepancy lies outside the error bars for this plot. By contrast, the model shows no such discrepancy for the rising envelope data set. In Figure 5, the plots of the predicted and observed excitation probabilities as a function of time from the heralding event, once more (not surprisingly) demonstrate the same asymmetry between the model and the results. Again, this is outside the quoted error bars for the decaying envelope measurements compared to the model predictions.

The source of this discrepancy could lie in the non-ideal transfer function of the Fabry-Perot cavity used to control the photon temporal envelopes, the behavior of the AOM switching on/off, motion of the atom in the trap as opposed to being stationary, and/or due to the light interacting with the atom having non-ideal polarization, for example. That the discrepancy depends on the shape of the photon envelope is remarkable. I would appreciate like some small discussion of the potential sources of this effect. [Also see d) below].

Reply: The theoretical curves shown in Figure 4 are indeed obtained from Eq.(5) by inserting Eq.(1) and Eq.(2). We clarified the corresponding description. The discrepancy between the theoretical and experimental absorption dynamics is most likely caused by the deviations of the photon envelopes to the ideal asymmetric functions (see reply to b).

Added paragraph to the main text (before Eq. 7):

'We obtain analytical solutions for the expected differences in transmission $\Delta(t)$ from Eq. 5 assuming the ideal photon envelopes Eq. (1-2). The magnitude and the dynamics of the observed scattering are well reproduced for $\tau_p = 13.3$ ns and $\Lambda = 0.033$ (Fig. 4, solid lines). The observed peak absorption for the exponentially decaying photon is slightly

higher than expected. We attribute this discrepancy to the imperfect photon envelopes which differ slightly from the ideal asymmetric exponential.'

c) The authors also say that "corrections for accidental coincidences were applied in the analyzed data shown in Fig. 4 and Fig. 5". I would appreciate a short explanation of these corrections here. The authors are taking the difference between two, almost equal traces to extract small, precise readings. The form of the correction and the method for correcting the data are relevant here.

Reply:

We clarify the applied corrections for accidental coincidences by refining our statements in the 'methods' section:

'Fig. 3 shows the coincidence histograms without additional processing. For the quantitative analysis (Fig. 4 and Fig. 5) we subtract the accidental coincidence rate from the histograms. The accidental coincidence rate is caused by background events in the photodetectors, and determined from the histograms by averaging the detected coincidences rate within a 300 ns wide time interval starting about 150 ns after the time interval used to analyze the scattering dynamics.'

d) In the text at the top of p.4, the authors state the percent differences between the excited state populations produced by the different shaped envelopes. The measured value shows a 56% larger peak excited state fraction as compared to the model prediction of 78% increase. This discrepancy should be commented upon in context with point b).

Reply: The differences between the peak excited state populations produced by the differently shaped envelopes is not as high as predicted. This is a consequence of the slightly higher than expected peak absorption rate of the exponentially decaying photon. We believe we have covered this discrepancy with the added paragraph under point (b).

e) Finally, the authors state: "The advantage of using exponentially rising photons is, therefore, to excite atoms at well defined instants in time." I suggest changing this to a more specific statement such as: "The advantage of using exponentially rising photons to excite a single atom is that this provides a larger peak excitation rate which is more temporally localized."

Reply: We clarified the statement mentioned by the reviewer and changed it to: 'The advantage of exciting single atoms with exponentially rising photons is a larger peak excitation probability within a narrower time interval.'

Overall, I enjoyed reading this manuscript: The authors present the results of carefully executed and thorough work. With some of the minor

adjustments I suggested above, I recommend publication of this work: The findings are of great interest to researchers in this field as well as to a general audience.

Reviewer #2:

The authors examine the excitation process of a single atom by a single photon. In particular, they investigate the influence of the shape of the wavepacket of the incoming photon, whether it is exponentially decaying or rising. The experimental results are valuable because some theoretical predictions had stated that efficient coupling of photons to atoms requires exponentially growing light fields to mimic the inverse of the spontaneous emission process.

The experiment is a continuation of a series of efforts in the lab of Prof. Kurtsiefer, where extinction of a focused laser beam by a single atom was studied. In 2013 they investigated the effect of the laser pulse shape on the excitation process. In 2014 they reported the production of single photons with exponentially rising profile. The current manuscript puts those two experiments together in a real tour de force effort, involving 1500 hours of data acquisition. The work would be a very nice addition to the literature in quantum optics. Therefore, I recommend this work for publication in Nature Communications once the following issues have been addressed:

1- The authors show convincingly the difference between the peak excitation probability for the cases of exponential rising and decaying pulses. They also state the important conclusion that when integrated over a long time interval both photon shapes are equally likely to be scattered. The latter statement should be included in the bold abstract (first paragraph) to avoid misleading/confusing the general reader about the necessity of pulse shaping when performing spectroscopy.

Reply: We included the following statement to the introductory paragraph:

'However, the overall scattering probability remains the same within the experimental uncertainties.'

2- In the second paragraph, the authors write "More recently, significant light-matter interaction has also been observed between single quantum systems and weak coherent fields in free space [18-21]." The authors seem to have selected references to vacuum experiments. Since the physics does not change in any way when the excitation is done in a dielectric, they should also include references to G. Wrigge, et al, Nature Physics 4, 60 (2008) and Vamivakas, et al. Nano Lett. 7, 2892 (2007) where this type of work was first demonstrated on molecules and quantum dots.

Reply: We added the two suggested references to the second paragraph.

3- In the last paragraph, the authors state "Our experimental results also contribute to a long-standing discussion about differences between

heralded and "true" single photons." They should cite the literature on this long-standing discussion.

Reply: We added four references covering arguments over the last two decades.

Reviewer #3:

1. Summary

The authors present their experiment on scattering a single photon off a single atom. The temporal shape of the input photon is set to be either exponentially rising or falling, depending on the detuning of a cavity. They demonstrate that the excited state probability as a function of time is different for the two cases, and that a higher probability is reached for the rising mode. The integrated probability is however the same, as predicted by the theory.

2. General comments

In general, I found the manuscript interesting and nicely written. The topic of the manuscript should be of interest to the general physics community. The authors show a nice agreement between experiment and the theory. The results are trustworthy and the statistical analysis (errorbars are shown and numbers are displayed with uncertainties) is ok. In conclusion, I find the manuscript appropriate for Nature Communications and I recommend publication.

Detailed comments:

3. Spatial overlap Λ

Can you discuss how you can improve the value for Λ ? For a Gaussian spatial mode, what is the best overlap with the atomic dipole pattern? Are there any ways to experimentally produce an input photon with $\Lambda=1$?

Reply: The spatial overlap can be improved by focusing the probe light to a smaller spot size. We included the following discussion in the main text of the paper which also includes one additional reference (Tey et al. New Journal of Physics 11 (2009) 043011) in the paragraph before the summary:

'The peak excitation probability and the signal rate can be improved by a larger spatial mode overlap Λ , which is currently limited by the numerical aperture of the focusing lens [31]. Other focusing geometries like parabolic mirrors can theoretically achieve complete mode matching $\Lambda = 1$ [23].'

4. Forward and backwards detectors:

Why do you have a better signal to noise ratio using the forward detector compared to the backwards detector? The atomic dipole emission should be the same in the forward and backwards directions, and I would therefore think that the number of detected photons (and thereby the signal to noise ratio) would be the same in the two directions.

Reply: Without considering the detector noise the forward and backward signal-to-noise ratios are indeed similar. The disadvantage of the signal at the backwards detector is that the signal rate is small compared to detector noise. We refined a statement in the main text after Eq. 4 regarding the signal at the backwards detector:
`However, the detection rate in such an experiment is relatively small and therefore susceptible to detector noise.'

5. y-label on Fig 3.

I am a bit confused about the y-label of figure 3. It says "Coincidence probability $\times 10^{-5}$ " and the y-axis goes from 10^{-5} to 10^{-3} . Does this mean that the actual probability goes from 10^{-10} to 10^{-8} ? To avoid confusion, I would suggest that you delete the " $\times 10^{-5}$ ", and just write out whatever the correct numbers are on the y-axis. Also, is it not clear to me how G_f and $G_{f,0}$ are normalized? What is the relation between the value of G_f and the time-bin duration of 2 ns?

Reply: The y-label of figure 3 was indeed incorrectly labelled. We corrected it and removed the " $\times 10^{-5}$ ". The G_f and $G_{f,0}$ are not normalized. Each 2-ns time-bin of the histogram represents the ratio of detected 'probe' photons to detected 'herald' photons.

6. Equation (7)

I am a bit puzzled about Eq. (7). The function $\Lambda(1-\Lambda)$ is maximal for $\Lambda=0.5$. Why is this Λ optimal? I would expect that $\Lambda=1$ would give the biggest epsilon. However, I see that $\epsilon=0$ for both $\Lambda=0$ and $\Lambda=1$. Please explain. Also, how can one make $\epsilon=1$? I see that $\epsilon=1$ for $\tau_p \gg \tau_0$. Is there a simple physical explanation why this is optimal? I would expect that $\tau_p = \tau_0$ would be optimal according to your argument about time-reversal.

Reply: The requirements for perfect excitation and perfect extinction ($\epsilon=1$) are different. Indeed, for perfect excitation when $\Lambda=1$, there is zero extinction. Consider that perfect excitation would require an incident photon covering the full solid angle of the atomic dipole emission pattern. The atom would be excited and eventually decay, and emit a photon in the same dipole pattern, i.e. the input mode. Thus the input photon comes back out in the same spatial mode, and the extinction is zero.

The case of perfect extinction has been considered, e.g. by Zumofen et al., PRL 101, 180404 (2008). Indeed, as the reviewer has pointed out, perfect extinction would require covering half the solid angle (i.e. $\Lambda=0.5$), and a photon with a sufficiently long coherence time ($\tau_p \gg \tau_0$).

7. 1500 hours

On the one hand, it is impressive that you can run your experiment for such a long time. On the other hand, it is not so nice that you need so long time to get good statistics. I think that it would be nice if you added some text in the methods section describing the main reasons for

needing so long measurement time and give some suggestions for how you can improve your setup in order to do faster measurements.

Reply: The long experimental data acquisition time was necessary because of the relatively low photon pair generation rate at a narrow bandwidth and low light+atom interaction strength due to the numerical aperture of the lens we used. We added a statement (see reply to #3 - 3.)

how the interaction strength can be improved by a larger spatial overlap Λ . Improvements of the photon pair generation rate are under way but we do not feel that the paper would benefit from a detailed discussion on this mostly technical effort.

8. Formatting of figure 2

Figure 2 has many details but is quite small. The figure would be more readable if it was larger and covered the full page width.

Reply: We agree and display Figure 2 with full page width. We hope that the editorial process can take care of this in an adequate way.

9. Formatting of Fig. 3, 4 and 5.

There is a problem with the formatting of Fig. 3, 4 and 5. The figure and the figure caption are partially on top of each other, making it hard to read the first two lines of each figure captions.

Reply: We seem to have no problems when compiling the LaTeX source with dvipdf or pdflatex and trust that the editorial conversion process at Nature Communications will take care of this.

REVIEWERS' COMMENTS:

Reviewer #1 (Remarks to the Author):

I would like to thank the authors for addressing the suggestions I submitting in my original review. Their responses were thorough and informative. With the changes made to the manuscript I recommend that this excellent research be published.

Reviewer #2 (Remarks to the Author):

The authors have responded to all comments in a satisfactory manner. The paper can be published as is.

Reviewer #3 (Remarks to the Author):

The reviewers, including myself, all recommended publication and had only minor comments. The authors have replied in a satisfactory way to those minor comments. I still recommend publication in Nature Communications.